# Fusion protein strategies for cryo-EM study of G protein-coupled receptors

Kaihua Zhang [1], Hao Wu[1,6], Nicholas Hoppe [2,6], Aashish Manglik [2,3,4] ✉ & Yifan Cheng [1,5] ✉

Single particle cryogenic-electron microscopy (cryo-EM) is used extensively to determine structures of activated G protein-coupled receptors (GPCRs) in complex with G proteins or arrestins. However, applying it to GPCRs without signaling proteins remains challenging because most receptors lack structural features in their soluble domains to facilitate image alignment. In GPCR crystallography, inserting a fusion protein between transmembrane helices 5 and 6 is a highly successful strategy for crystallization. Although a similar strategy has the potential to broadly facilitate cryo-EM structure determination of GPCRs alone without signaling protein, the critical determinants that make this approach successful are not yet clear. Here, we address this shortcoming by exploring different fusion protein designs, which lead to structures of antagonist bound $A_{2A}$ adenosine receptor at 3.4 Å resolution and unliganded Smoothened at 3.7 Å resolution. The fusion strategies explored here are likely applicable to cryo-EM interrogation of other GPCRs and small integral membrane proteins.

G protein-coupled receptors (GPCRs) are the largest family of integral membrane proteins. All GPCRs share a common seven-transmembrane helices (7TM) architecture and are divided into six families, from A to F[1]. Upon agonist binding, GPCRs activate a wide variety of downstream signaling pathways by interacting with heterotrimeric G proteins, G protein-coupled receptor kinases (GRKs), and arrestins[2]. As a family, GPCRs represent an important class of drug targets for many human diseases[3]. Over the past decade, methodological advances have provided deep insights into the structure and mechanism of GPCR function. Continued structural interrogation of GPCRs in all functional states, including unliganded apo, inactivated by antagonists, activated by agonists, and in complexes with downstream signaling proteins, remains critically important in understanding GPCR signal transduction as well as facilitating therapeutic development.

X-ray crystallography initially provided a robust method to determine many GPCR structures. Although integral membrane proteins like GPCRs are challenging to crystallize, a highly successful strategy is to engineer a fusion protein replacing the third intracellular loop (ICL3), located between transmembrane helices 5 (TM5) and 6 (TM6), so that the inserted fusion protein can mediate crystal contacts[4,5]. This approach, combined with lipidic cubic phase crystallography, has led to over 400 deposited structures in the RCSB Protein Databank. While X-ray crystallography has been highly successful for obtaining structures of GPCRs bound to either high-affinity antagonists or agonists, obtaining structures of unliganded receptors or GPCR-signaling protein complexes has remained challenging, likely due to structural dynamics of unliganded or active-state receptors that inhibit crystal formation.

Single-particle cryogenic-electron microscopy (cryo-EM) has emerged as a revolutionary approach to structure determination. Instead of relying on protein crystallization, structure determination by cryo-EM relies on computationally aligning and averaging images of

[1]Department of Biochemistry and Biophysics, University of California San Francisco, San Francisco, CA 94158, USA. [2]Department of Pharmaceutical Chemistry, University of California San Francisco, San Francisco, CA 94158, USA. [3]Department of Anesthesia and Perioperative Care, University of California San Francisco, San Francisco, CA 94158, USA. [4]Chan Zuckerberg Biohub, San Francisco, CA 94158, USA. [5]Howard Hughes Medical Institute, University of California San Francisco, San Francisco, CA 94158, USA. [6]These authors contributed equally: Hao Wu, Nicholas Hoppe. ✉e-mail: Aashish.Manglik@ucsf.edu; Yifan.Cheng@ucsf.edu

individual molecules to yield a three-dimensional (3D) reconstruction[6,7]. A central challenge in cryo-EM, however, is that the resolution of a 3D reconstruction relies on the accuracy of aligning images of individual molecules with each other. Most GPCRs have an average molecular weight smaller than 40 kDa with critical structural features embedded within the membrane; in purified systems, these regions often exist within a detergent micelle or a lipid nanodisc. For such small membrane proteins without clear distinguishable structural features protruding from 7TM domain, particle alignment is exceptionally challenging. Nonetheless, single-particle cryo-EM has revolutionized the structure determination of GPCR-signaling protein complexes. A key reason for this success is that the rigidly attached signaling proteins drive particle alignment of the 7TM region within the membrane (Supplementary Fig. 1a). More broadly, for small membrane protein targets without distinct structural features outside of the membrane, adding a fiducial marker like an antibody fragment (Fab) can facilitate accurate image alignment if rigidly associated[8].

In an ideal setting, the single-particle cryo-EM approach would also enable routine structure determination of GPCRs in the absence of signaling proteins. This would potentially enable interrogation of antagonist-bound receptors or, particularly, unliganded receptors without the challenges inherent in crystallization. While there are now a few cryo-EM structures of unliganded family B and C GPCRs, these receptors form stable dimers in vitro and/or have relatively stable extracellular domains (Supplementary Fig. 1b)[9–11]. More recently, nanobody has been demonstrated to enable the high-resolution reconstructions of GPCRs in an inactive state by recognizing a grafted intracellular loop[12]. In principle, rigid attachment of a fusion protein to a smaller GPCR without a large extracellular domain may be sufficient to drive particle alignment in the membrane region, yielding a high-resolution reconstruction of the 7TM domain. Indeed, recent success with the family F human Frizzled5 (hFzd5) receptor demonstrated that an ICL3 fusion of the apocytochrome b562 fusion protein BRIL, combined with an anti-BRIL Fab fragment and an anti-Fab nanobody, could yield an interpretable reconstruction of the receptor 7TM domain (Supplementary Fig. 1c)[13]. A fundamental limitation, however, is that the design principles required for such successful reconstructions remain unclear.

Our goal in this study is to understand what factors enable successful 3D reconstruction of individual GPCRs using a fusion protein strategy. We aim to address two important design considerations: (1) is there a specific fusion protein size that is required for successful particle alignment, and (2), is the rigid attachment of a fusion protein by helix extension required for particle alignment? We use two model systems for this interrogation, the adenosine $A_{2A}$ receptor ($A_{2A}R$) and the Smoothened receptor (SMO)[14–22], both of which have previously been interrogated by X-ray crystallography and cryo-EM. By exploring various fusion strategies, we determine a 3.4 Å resolution structure of antagonist-bound inactive $A_{2A}R$ in detergent micelle and a 3.7 Å structure of unliganded inactive SMO in a lipidic environment. These two examples provide design guidelines for use of fusion proteins to enable single-particle cryo-EM that are likely applicable to other GPCRs and other small integral membrane proteins.

## Results

### $A_{2A}R$ structure enabled by a rigidly attached fusion protein bound to a Fab

We first explored whether a GPCR fusion protein strategy could enable high-resolution structure determination of the 7TM region. Encouraged by a recent cryo-EM structure of hFzd5 at 3.7 Å resolution (Supplementary Fig. 1c)[13,23], we sought to identify a "minimal requirement" for the fusion protein strategy. We therefore used a construct of the $A_{2A}R$ that previously yielded a 1.8 Å crystal structure (Supplementary Fig. 1d)[24]. Several features of this construct are notable. First, the structure was determined with the high-affinity antagonist ZM241385.

Second, this construct contains thermostabilizing mutations that likely further limit conformational heterogeneity. Finally, the crystal structure of this construct revealed that the BRIL domain is rigidly linked to the receptor 7TM domain by two continuous helices connecting TM5 and TM6 of the receptor to the N- and C- terminus of BRIL, respectively. These features make it an ideal example to test the requirement of a fusion protein to drive image alignment. We therefore purified $A_{2A}R$-BRIL-fusion construct[24] in L-MNG/CHS bound with ZM241385 for cryo-EM studies (Supplementary Fig. 2).

We initially attempted to determine a structure of ZM241385 bound $A_{2A}R$-BRIL by relying on the rigidly attached BRIL domain (MW ~10 kD) as the sole fiducial. Despite extensive data collection and image processing, we were unable to either identify clear 2D class averages with sufficient clear structural features (Supplementary Fig. 2a, b), or generate a reasonable 3D reconstruction, even when we used the low pass filtered crystal structure as an initial reference model. These results suggest that a BRIL domain alone, even rigidly attached to the protein, does not provide sufficient features to enable particle alignment of a GPCR. Following the successful strategy employed with hFzd5, we next added an anti-BRIL Fab fragment to enlarge the fiducial marker. Apparently, the larger fiducial marker successfully drives particle alignment and yielded a reconstruction with a global resolution of ~3.4 Å. Further focused refinement improved the resolution of the transmembrane domain to 3.2 Å, sufficient to enable model building (Fig. 1, Supplementary Figs. 2 and 3).

The cryo-EM reconstruction of $A_{2A}R$-BRIL/Fab is almost identical to the X-ray crystal structure of $A_{2A}R$-BRIL, with a global root mean squared deviation (RMSD) of 0.9 Å. The ligand ZM241385 is well resolved in the ligand-binding pocket (Fig. 1d, e), demonstrating that this strategy can enable visualization of drug binding. Furthermore, we identified a lipid density with two aliphatic tails and a polar headgroup between TM5 and TM6 on the inner leaflet of the membrane adjacent to $A_{2A}R$, which was not observed in any of the previous 58 X-ray and cryo-EM structures of $A_{2A}R$ in the Protein Data Bank. We putatively assigned it as phosphatidylserine, as it matches best with the density (Fig. 1f) and is consistent with native mass spectrometry studies identifying phosphatidylserine as a co-purifying lipid with $A_{2A}R$[25]. Based on these results, we conclude that a Fab tightly bound to a rigidly attached BRIL is necessary to serve as a fiducial marker for single-particle image alignment.

### A single helix BRIL connection is insufficient for high-resolution reconstruction

We next aimed to understand what factors drive the rigid attachment of a fusion protein to a GPCR. For both $A_{2A}R$ and hFzd5, BRIL was attached to the 7TM domain with two extended helices. This arrangement, however, requires that the orientations of TM5 and TM6 match precisely with the N- and C-terminal helices in BRIL; indeed, the hFzd5 structure required significant engineering at this junction for success[13]. More commonly, the BRIL domain has been inserted into a GPCR or other target protein only with a single extended helix[19], as rigid attachment is not a requirement for crystallization. We thus asked whether a fusion protein attached to a target protein with only an extended helix at one linker site can facilitate high-resolution structure determination when the other is not necessarily grafted with a rigid or short linker.

For this approach, we used mouse SMO (mSMO), another GPCR with previously determined crystal and cryo-EM structures. Wildtype mSMO contains a 7TM domain and an extracellular cysteine-rich domain (CRD). Previous cryo-EM studies of SMO bound to $G_i$ were unable to resolve the CRD region[17,18], suggesting that it is flexible and unlikely to drive image alignment. However, like $A_{2A}R$, insertion of a BRIL domain in the SMO ICL3 facilitated SMO crystallization (Supplementary Fig. 1e)[14,19]. Notably, unlike the $A_{2A}R$ crystal structure, this construct has only a single extended

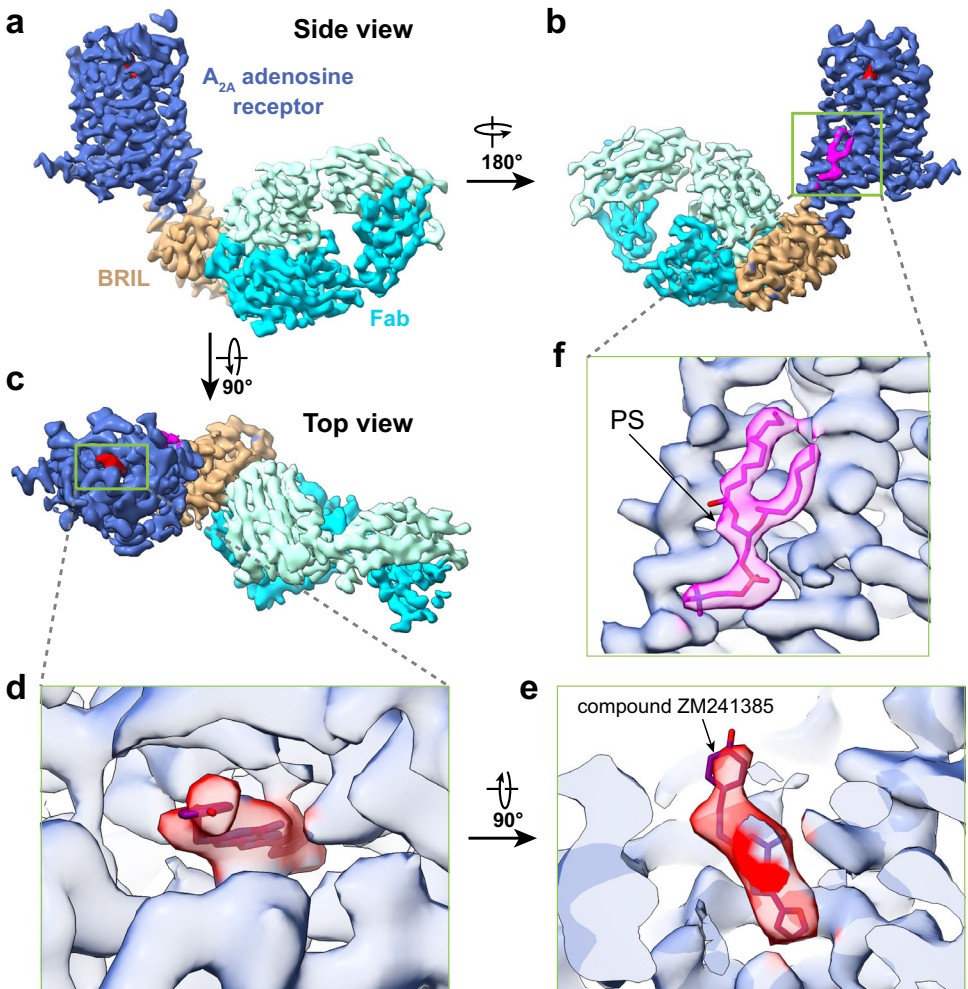

**Fig. 1 | Single-particle cryo-EM structure of A₂ₐR-BRIL. a–c** Three different views of cryo-EM density map of A₂ₐR-BRIL bound with an anti-BRIL Fab. **d, e** Close-up views of compound ZM241385 in the binding pocket. **f** The location of lipid density highlighted by the docking of the phosphatidylserine.

helix between TM5 of SMO and the N-terminus of BRIL. We therefore used this fusion strategy to test whether BRIL attachment with a single extended helix, with or without an anti-BRIL Fab, is sufficient to drive image alignment. Early studies have used the lipid nanodisc as a controlled membrane environment to assay the requirement of cholesterol for SMO activity[26]. We thus reconstituted the purified mSMO-BRIL into lipid nanoparticles formed by saposin (Salipro)[27] with added cholesterol. As a membrane scaffolding protein, saposin allows the formation of lipid nanoparticles that are adaptive to the size of the reconstituted membrane proteins, providing the benefit of a lipid environment but without adding excessive unstructured density to the nanoparticles. Consistently, as in the case of A₂ₐR-BRIL without an anti-BRIL Fab, we were unable to determine a high-resolution structure of mSMO-BRIL alone (Supplementary Fig. 5a, d). However, unlike in the case of A₂ₐR-BRIL/Fab, the addition of the anti-BRIL Fab to enlarge the fiducial marker did not yield any improvement (Supplementary Fig. 5e). This is unlikely caused by the reconstitution of mSMO into saposin nanoparticles (see below). We surmised that either a BRIL alone is insufficiently large to drive image alignment (as shown above in the case of A₂ₐR-BRIL) or the attachment of a BRIL alone with a single extended helix is likely insufficiently rigid, and the flexibility is amplified by the addition of an anti-BRIL Fab. These results indicate that a construct with two extended helices to BRIL bound with an anti-BRIL fab is likely important to achieve sufficient resolution in cryo-EM reconstructions.

### Structure of apo SMO in a lipid nanodisc enabled by PGS fusion

We turned to another fusion protein approach to enable structure determination of inactive apo mSMO. Given the challenges in precisely inserting a fusion protein with two extended helices, we considered that attaching a larger fusion protein via a single rigid helix may be easier to design. We chose a thermostable glycogen synthase domain from *Pyrococcus abyssi* (PGS) (MW ~20 kD), which is a fusion protein that has been used to determine GPCR crystal structures, including the CB1 cannabinoid receptor[28], and the OX1 and OX2 orexin receptors[29] (Supplementary Fig. 1f). We anticipated that the PGS alone, which is bulkier than BRIL but smaller than BRIL/anti-BRIL fab, may be sufficient to function as a fiducial marker with a single extended helix, assuming a rigid attachment.

We designed two different constructs, mSMO-PGS1, in which the PGS domain is inserted to SMO after position 441 on ICL3 and before position 445 on TM6, and mSMO-PGS2, in which PGS has inserted one and a half helical turns (five amino acids) up towards in TM6 (Supplementary Fig. 4). AlphaFold[30] predicts that a PGS C-terminal helix is extended to TM6 of mSMO as a continuous helix in both constructs but with PGS oriented to the side of the 7TM bundle in mSMO-PGS1 and directly under the 7TM bundle in mSMO-PGS2 (blue and yellow ribbons in Fig. 2a). In both cases, N-terminus of PGS is connected to the C-terminal end of TM5 via a loop. Same as above, we reconstituted the purified mSMO-PGS into lipid saposin nanoparticle[27] with added cholesterol. We obtained well-behaved mSMO-PGS protein reconstituted in nanoparticles and prepared cryo-EM grids using the peak fraction

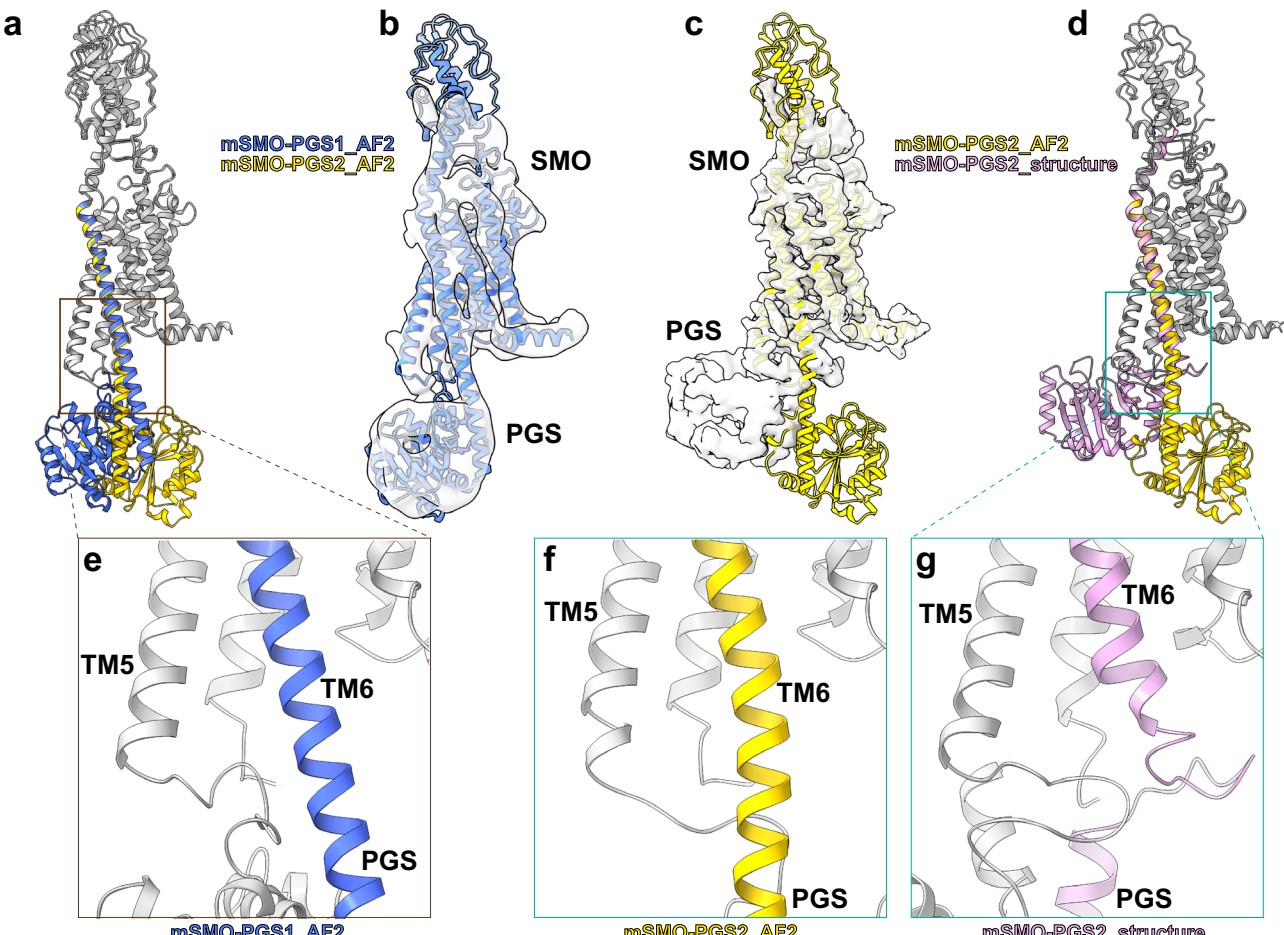

**Fig. 2 | Single-particle cryo-EM structures of mSMO-PGS. a** The atomic models of mSMO-PGS1 (blue) and mSMO-PGS2 (yellow) predicted by alphaFold2 (AF2). Note that the inserted PGS are predicted to be oriented in two opposite directions in these two constructs. **b** Cryo-EM density determined from mSMO-PGS1, docked with the atomic model predicted by AF2 (blue ribbon). **c** Cryo-EM density determined from mSMO-PGS2, docked with the atomic model predicted by AF2 (yellow ribbon). **d** Overlay of determined structure (pink ribbon) and predicted atomic model for mSMO-PGS2. **e**–**g** Enlarged views of connection region between SMO TM6 and PGS in the predicted model of mSMO-PGS1 (**e**), predicted model of mSMO-PGS2 (**f**), and the determined structure of mSMO-PGS2 (**g**).

from size-exclusion chromatography (Supplementary Fig. 5b-c, and f-g).

We successfully determined cryo-EM structures from both mSMO-PGS constructs. These structures varied greatly in a resolution despite small differences in these constructs (Fig. 2, Supplementary Figs. 6–8). We only achieved ~6 Å resolution from mSMO-PGS1 (Fig. 2b and Supplementary Fig. 6), which revealed an overall architecture of the fusion protein consistent with the AlphaFold Prediction. Although insufficient for de novo model building, the resolution is sufficient to allow the placement of TM bundles. The orientation of the TM6 and PGS indicates that the connection between mSMO and PGS maintained an extended helix, matching the AlphaFold prediction. This type of connection is similar as how PGS is connected CB1 cannabinoid receptor[28], and the OX1 and OX2 orexin receptors[29], as revealed by the crystal structures of these receptors.

We concluded that similar to the case of mSMO-BRIL construct, single helix extension to PGS fusion is insufficient for high-resolution reconstruction.

The reconstruction of mSMO-PGS2 is, surprisingly, different from the AlphaFold prediction but reaches a significantly better resolution of 3.7 Å (Fig. 2c, Supplementary Figs. 7 and 8). Unlike the AlphaFold prediction, the PGS is oriented beneath TM bundle without a continuous helix between TM6 of mSMO and the C terminus of PGS, but a contorted loop that allows the PGS domain to directly interact with the mSMO 7TM bundle. At the interface of

mSMO and PGS, a hydrophobic loop of PGS (F1124, L1126, L1129, I1149, and F1195) contacts hydrophobic residues at the base of TM3 and TM5 (F347, L350, I433, and L440) (Fig. 3a–c). This unpredicted hydrophobic interaction with the intracellular side of mSMO likely stabilized the orientation of PGS relative to mSMO. We surmised that this interaction enabled a rigid attachment of PGS to mSMO and facilitated a higher resolution reconstruction of the mSMO 7TM domain. It is likely similar interaction can occur in other GPCRs if a PGS domain is inserted as a fusion protein with two nonrigid loop connections to TM5 and TM6. Indeed, we found two examples, in which fused PGS facilitated crystallization of MC4R (PDB code: 6W25[31]) and MT₁ (PDB code: 6ME2[32]). Both crystal structures show similar hydrophobic interactions between PGS domain and GPCRs (Fig. 3d–i), and the buried areas between PGS and GPCRs calculated from these three structures are 836 Å$^2$ (mSMO), 350 Å$^2$ (MC4R), and 533 Å$^2$ (MT₁).

Similar hydrophobic residues can be also found in H$_1$R and GLP-1R (Supplementary Fig. 9a, b). Notably, the equivalent hydrophobic residues in human SMO also interact with the hydrophobic residues in α5 helix of G$_i$ in the SMO-G$_i$ complex (PDB ID: 6XBM) (Supplementary Fig. 9c), suggesting that PGS and G$_i$ interact with SMO similarly, even though SMO is in an unliganded inactive state in this study but in agonist bound activated state in complex with G$_i$. Considering that these highly conserved hydrophobic residues in G proteins (Supplementary Fig. 9d–f) are seen to interact with the similar hydrophobic

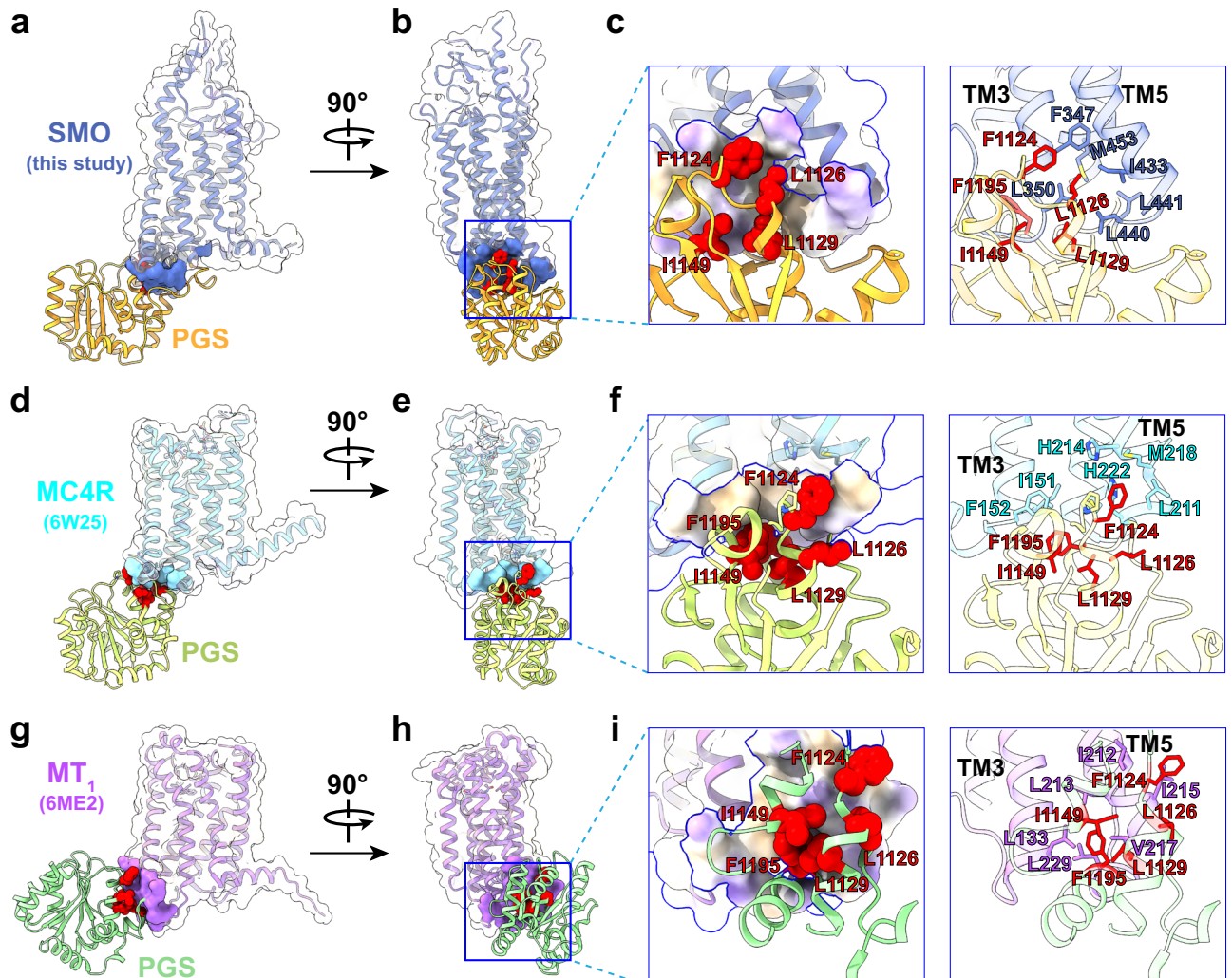

**Fig. 3 | Hydrophobic interfaces between SMO/MC4R/MT1 and the PGS fusion protein. a, b** Two different views of mSMO-PGS2 structure showing the relative orientation of the receptor and PGS. The contact area of SMO with PGS is indicated by blue surface. **c** An enlarged view of the hydrophobic interactions between mSMO and PGS that facilitate the rigid attachment of PGS, including the key residues in PGS (as sphere on the left) and SMO, respectively. The contact area is colored to show the hydrophobicity. **d, e** Two different views of MC4R structure

(PDB ID: 6W25 ref. 31). The contact area of MC4R with PGS is indicated by cyan surface. **f** The enlarged views of the boxed regions show the interactions of the same PGS residues as observed in mSMO-PGS2 with nonpolar residues in MC4R. **g, h** Two different views of MT₁ structure (PDB ID: 6ME2 ref. 32). The contact area of MT₁ with PGS is indicated by aquamarine surface. **i** The enlarged views of the boxed regions show the interactions of the same PGS residues as observed in mSMO-PGS2 with nonpolar residues in MT₁.

area of other GPCRs, we predict that a similar interaction between PGS and the TM3/TM5 region of other GPCRs is possible.

A comparison of the two structures we determined here suggests that PGS as a fusion protein is sufficient to facilitate high-resolution structure determination but requires a stable attachment to SMO. Relatedly, a single extended helical connection to PGS without additional protein–protein interaction is insufficient to produce the required structural rigidity. This conclusion is further substantiated by our earlier structurally inconclusive results using a single-helical mSMO-BRIL fusion. In contrast, the hydrophobic interaction between PGS and mSMO in mSMO-PGS2 provides sufficient stabilization to achieve high-resolution structures (Fig. 3c and Supplementary Fig. 9a).

### The structure of apo SMO in a lipidic environment reveals a sterol site

SMO is a key component of the Hedgehog signaling pathway, implicated in embryonic development and adult tissue homeostasis[33]. SMO dysfunction leads to birth defects and cancer[34]. SMO activity is regulated by Patched, a receptor of Hedgehog (Hh). A current model for

Hedgehog signaling suggests that Patched transports sterols across the membrane and maintain a low local sterol concentration in the membrane inner leaflet; this keeps SMO inactive[35]. Inhibition of Patched upon Hedgehog binding increases inner leaflet sterol concentration and releases SMO inhibition. Among seventeen published SMO structures to date, there are 12 crystal structures of the receptor alone or in a complex with a nanobody and five cryo-EM structures of the receptor in complexes with $G_i$ proteins, although the participation of $G_i$ protein in SMO signaling pathway remains controversial[36].

Our cryo-EM structure of mSMO in saposin nanodisc enabled modeling of the 7TM and linker domain (LD), which connects the extracellular CRD with the 7TM domain (Fig. 2c). The bulk of the SMO CRD is poorly resolved in our map, except for two small density fragments in the region connecting the 7TM to the CRD, one of which has the appearance of a short helix. Structural flexibility of the CRD in apo SMO is similar to previous cryo-EM structures of active SMO bound to $G_i$, in which the CRD is unresolved. By contrast, a previous crystal structure of SMO alone resolved the CRD, suggesting that this particular orientation is likely stabilized by crystal contacts. We docked three available atomic models of SMO with CRD, each representing

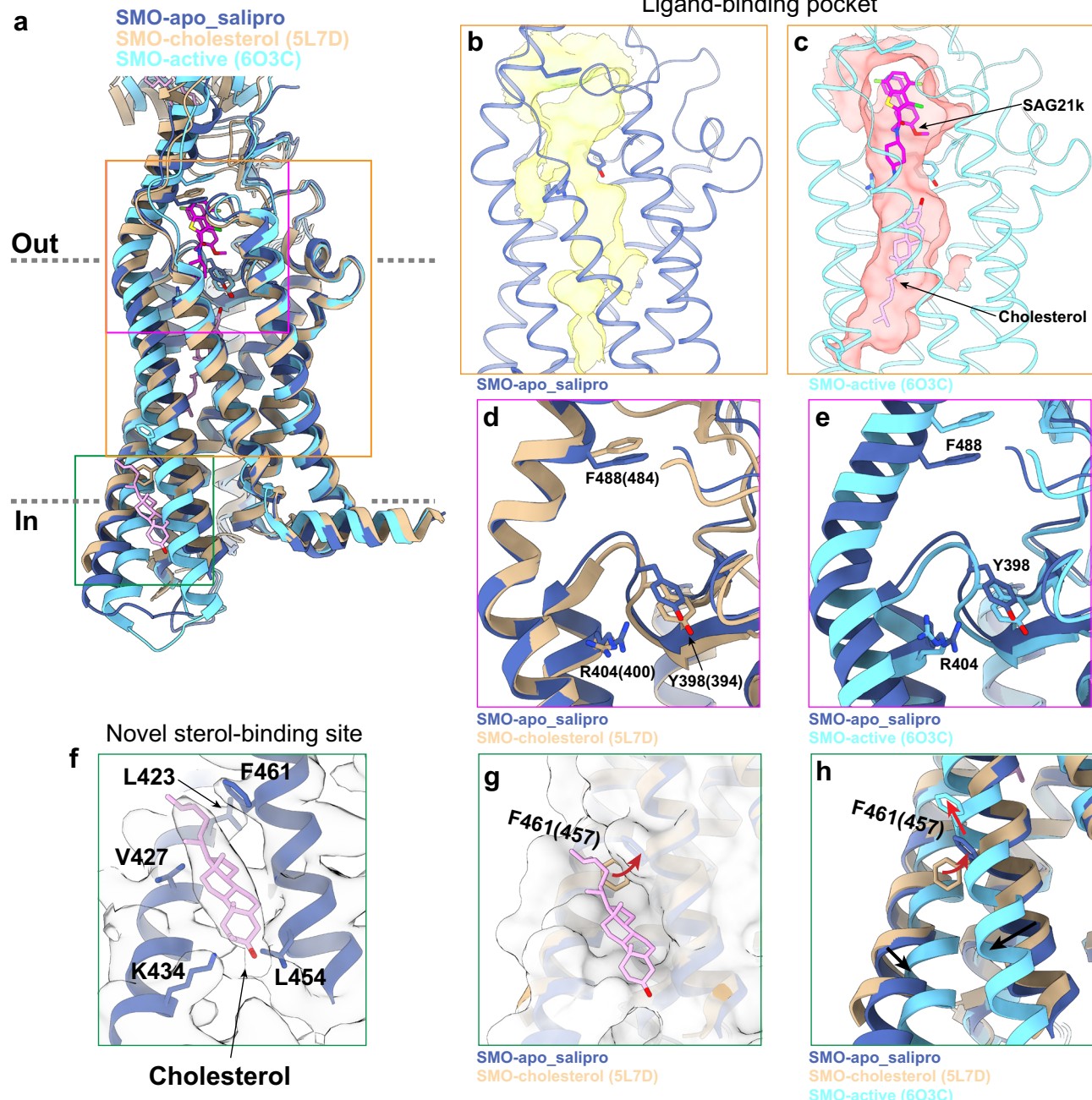

**Fig. 4 | Structure of mSMO in the apo state. a** Comparisons of structures of mSMO in apo state (this study) with antagonist-bound inactive (PDB ID: 5L7D) and agonist bound active (PDB ID: 6O3C) states. Three structures are color labeled. **b** The ligand-binding pocket calculated from the apo state structure determined in this study. The pocket is seen empty with space insufficient to accommodate either agonist or cholesterol. **c** In comparison, the ligand-binding pocket of SMO in the active state (PDB ID: 6O3C) with the agonist SAG21k bound with a cholesterol seen in the bottom of the pocket. **d** Overlay of the atomic models of SMO apo structure determined in this study (blue ribbon) with the one determined previously (PDB ID: 5L7D) without the ligand but with a bound cholesterol in CRD. The noticeable differences between the two suggest that, although both structures are in apo states, they may present two different intermediate states of apo SMO. **e** Overlay of the structures shown in **b**, **c**. **f** The density of a cholesterol located in the inner leaflet side of mSMO. **g** The sterol site highlighted by surface view. **h** The same view as seen in **g** to show the effect of conformational change from inactive to active states on sterol binding.

different orientations of the CRD relative to 7TM bundle, into our density map. The short helix-like density does not match any helices in the docked CRD domain (Supplementary Fig. 10), suggesting that the conformation of CRD in our structure may be different from all other available structures, or, alternatively, that CRD is more flexible in the absence of ligands or crystal packing.

In comparison with previous structures of SMO, our structure of apo SMO resembles the inactive SMO, with an overall RMSD of 0.75 Å in the 7TM region and a conformation of TM5 and TM6 that is

consistent with an inactive receptor (Fig. 4a). SMO contains a large pocket within the 7TM domain, which is the binding site of endogenous sterols important in pathway activation and synthetic small molecule agonists, antagonists, and allosteric modulators[15,19]. Consistent with an apo state, this pocket is empty in our structure without the obvious density of any bound ligand or sterols (Fig. 4b). Indeed, the ligand-binding pocket in apo SMO is too narrow to accommodate a sterol within the 7TM bundle as observed in one of previous active SMO structures (PDB ID: 6O3C)[15] (Fig. 4b, c).

We compared our structure of lipid-embedded apo SMO with a previously determined X-ray crystal structure of unliganded SMO solved using the lipidic cubic phase method (PDB ID: 5L7D)[14]. This previous structure resolved the CRD bound to cholesterol, but with no clear cholesterol density within the 7TM. Compared to this structure and the active-state structure of SMO (PDB ID: 6O3C), we identified subtle, but important differences in the 7TM region. First, several residues in the 7TM pocket rearrange, including Y398 (Y394 in human SMO (hSMO)), R404 (R400 in hSMO) and F488 (F484 in hSMO) (Fig. 4d, e and Supplementary Fig. 8e). This leads to a smaller pocket volume (1630 vs 2292 Å$^3$). Second, we identified a density adjacent to TM5 and TM6 outside the 7TM bundle facing the inner leaflet of the plasma membrane that is consistent in shape and size with a sterol (Fig. 4f). The resolution precludes assignment of a specific sterol, and it is unclear whether this density represents an endogenous sterol or cholesterol added during saposin reconstitution. However, given the high concentration of cholesterol used in our reconstitution, we have tentatively modeled cholesterol at this site. To our knowledge, no lipid-like density at a similar site has previously been observed in X-ray or cryo-EM structures of SMO. The modeled cholesterol binds in a hydrophobic crevice formed by residues L423 and V427 in TM5, and L454 and F461 in TM6. The side chain of residue K434 in TM5 may interact with the hydroxyl group of cholesterol (Fig. 4f). Notably, F461 (F457 in hSMO) reorients to accommodate binding of cholesterol at this site (Fig. 4g). Comparison with the active-state structure of SMO shows that this crevice is disrupted upon receptor activation (Fig. 4a and h), suggesting that this site only exists in the inactive or preactive state[15]. Finally, the iso-octyl tail of cholesterol faces the opening of a tunnel between the inner leaflet of the lipid bilayer and the 7TM pocket of active SMO (Fig. 4a, c).

These structural observations lead us to speculate that sterol binding at this site may be important in Hedgehog pathway function. We propose that cholesterol initially binds at this site in SMO upon pathway activation. Further conformational changes in TM5 and TM6 squeeze out cholesterol at this site so that it can flip and reach the cholesterol-binding site within the TM bundle by entering an opening tunnel between TM5 and TM6. While speculative, our proposal extends the emerging model that inner leaflet cholesterol level is regulated by Patched-1, and that pathway activation leads to an increase in inner leaflet cholesterol that is sensed by SMO[15–17,35].

## Discussion

The application of cryo-EM to GPCR structural biology has greatly accelerated the determination of GPCR-signaling protein complex structures. Yet determining structures of noncomplexed receptors remains difficult because of the inability to align individual particles. Drawing inspiration from the use of fusion proteins in crystallography and in the structure determination of hFzd5 by cryo-EM, we set out to interrogate existing strategies and establish new fusion strategies for GPCR cryo-EM structures. It is worth mentioning that the purpose of adding a fusion protein to GPCRs is only to provide a fiducial marker to facilitate image alignment, but not to reduce protein dynamics or improve stability, which also influences the achievable resolutions of cryo-EM structures. We thus focus on GPCRs that have been stabilized by mutations or antagonists.

We began by testing the BRIL-fusion strategy on two model GPCRs, A$_{2A}$R and SMO. Previously solved X-ray crystallography structures of BRIL fusion constructs showed that the A$_{2A}$R-BRIL forms two continuous helices between the receptor and BRIL, while SMO-BRIL only forms a single continuous helix. For A$_{2A}$R, the use of an anti-BRIL Fab enabled image alignment yielding a high-resolution reconstruction of the receptor and its drug binding pocket. However, particle alignment failed without the Fab. For SMO, neither the BRIL fusion alone nor the BRIL fusion plus anti-BRIL Fab enabled a high-resolution reconstruction. Taken together, our results show that for GPCRs

without large and stable extracellular domains, the BRIL-fusion approach requires both rigid linkers between BRIL and a receptor, such as with two continuous helices or one continuous helix supported by another optimized linker, as demonstrated in a recent study of EBI2[37,38], and an additional Fab to enlarge the size of the fiducial marker.

After identifying the constraints of the BRIL-fusion strategy, we attempted to test a new fusion strategy with less inherent constraints on helical fusion points and using different fiducial markers. We chose to incorporate PGS into ICL3 of mSMO, hypothesizing that the structural addition of a single continuous helix and the size of PGS would be sufficient to facilitate image alignment. Surprisingly, we learned that a single continuous helix is still insufficient. Instead, an extension without helical character allows stable hydrophobic contacts between the SMO and PGS, which independently enabled particle alignment and eventual reconstruction of the SMO 7TM domain to 3.7 Å resolution. This finding presents an alternative strategy for determining GPCR structures by cryo-EM.

Our structures of A$_{2A}$R and mSMO provided structural insights into how lipids interact with these two receptors. For A$_{2A}$R, our cryo-EM structure revealed putative phosphatidylserine bound to the receptor between TM5 and TM6. This lipid has not been observed in any previously solved X-ray structures of A$_{2A}$R. For SMO, our structure of the inactive receptors in a lipid environment revealed subtle, but important conformational differences from previously published X-ray crystal and cryo-EM structures. Furthermore, this structure revealed a sterol binding site between TM5 and TM6. Both cases highlight the potential utility of cryo-EM in illuminating aspects of GPCR structural biology.

In addition to the recent nanobody strategy by recognizing a grafted intracellular loop[12], our studies highlight the importance of rigid coupling between the GPCR and fusion protein for structure determination by cryo-EM. We demonstrate two alternative approaches to achieve such rigidity. One is to attach the fusion protein to GPCRs with two extended helices, typically TM5 and TM6. A single extended helix at one linker site connecting the GPCR to the fusion protein does not likely yield sufficient rigidity to drive image alignment for high-resolution structure determination, particularly when the other linker is not rigid but long and/or flexible. If TM5 and TM6 can both be connected directly to the fusion protein as extended helices, as in the case of A$_{2A}$R-BRIL, the insertion is likely sufficiently rigid for structure determination. AlphaFold structure prediction may further enable such designs. An alternative, unexpected approach, is that a fusion protein may interact with a GPCR stably by other specific interactions, such as the hydrophobic interaction between PGS and mSMO. In this case, the PGS alone is of sufficient size to drive image alignment. Notably, the hydrophobic residues in mSMO interacting with PGS are conserved in many other GPCRs, suggesting that PGS fusion protein could be more broadly used for structural studies of other GPCRs. Interestingly, mSMO-PGS1 and mSMO-PGS2 represent two types of configurations of PGS fusion domain relative to the TM domain of GPCRs, both of which are seen in multiple previously published crystal structures of GPCRs[28,29,31,32]. Our study presented here shows that only one of the two can facilitate image alignment. Therefore, when applying this approach to other GPCRs, it is necessary to try multiple designs of the linkers to obtain the desired one. Our explorations of fusion protein strategies therefore enable structural determination of GPCRs without co-complexed signaling proteins, thereby opening new avenues for understanding GPCR structure, function, and eventual drug design.

## Methods

### SMO expression, purification, and Salipro reconstitution

Recombinant mouse SMO gene (amino acids 64–566) was cloned into a pFastBac1-CMV vector containing an expression cassette with

an HA signal sequence followed by a Flag epitope tag (sequence: DYKDDDDA), Strep-tag II peptide (sequence: WSHPQFEK), and tobacco etch virus (TEV) protease recognition site at the N-terminus, with PGS or BRIL fused in ICL3 and with a rhinovirus 3 C protease recognition site and 8xHis tag at the C terminus of SMO. Through the BacMam approach, Sf9 insect cells and HEK293S GnTI⁻ cells were used to generate the baculovirus and 48 h expression of SMO with 10 mM sodium butyrate added after 24 h transduction, respectively. Cells were collected by moderate centrifugation and stored at −80 °C until further use. Frozen cell pellets were thawed in the buffer containing 25 mM HEPES, pH 7.4, 150 mM NaCl, and 400 uM TCEP, supplemented with protease inhibitor cocktail (sigma), and then solubilized with the working buffer containing 40 mM HEPES, pH 7.4, 150 mM NaCl, 0.75% (w/v) n-dodecyl-β-D-maltopyranoside (DDM, Anatrace)/0.15% (w/v) cholesteryl hemisuccinate (CHS, Sigma), 10% glycerol, and 400 uM TCEP for 2.5 h at 4 °C. After centrifugation, the resulting supernatant was loaded on Ni-NTA resin (Thermo Scientific) which was subsequently treated with 50 mM HEPES, pH 7.4, 150 mM NaCl, 0.03% DDM/0.006% CHS, 10% glycerol, and 100 uM TCEP plus 30 mM imidazole for washing and plus 200 mM imidazole for eluting, respectively. After incubation with ANTI-FLAG M2 Affinity Gel (sigma) overnight at 4 °C, the beads were washed extensively and the receptors were eluted with 50 mM HEPES, pH 7.4, 150 mM NaCl, 0.03% DDM/0.006% CHS, 10% glycerol, 100 uM TCEP, and 0.2 mg/ml 3xFlag peptide.

For Salipro reconstitutions of both SMO-PGS and SMO-BRIL samples, membrane scaffold protein Saposin A was expressed and purified from *Escherichia coli*. Lipids mixture (55.2%POPC:36.8% POPG:8%cholesterol at mass ratio) was prepared as previously described[39]. Purified SMO protein in detergent (DDM/CHS) was mixed with Saposin A and lipids mixture using the malor ratio SMO:Saposin A:lipids = 1:3:47 and incubated on ice for 10-30 min. Detergents were removed by adding 2–3 batches of Bio-beads SM2 (Bio-Rad) with constant rotation overnight. The reconstituted receptors were cleared by moderate centrifugation via spin filter microtubes (Merck Millipore Ltd.) followed by size-exclusion chromatography using a Superdex 200 Increase 10/300 GL column (GE) equilibrated with buffer containing 20 mM HEPES-NaOH, pH 7.4, 150 mM NaCl, and 100 μM TCEP. The peak fractions corresponding to SMO in Salipro particles were pooled, concentrated to around 2.6 mg ml⁻¹ using Amicon Ultra filter device (50 kDa MWCO, Millipore), and assessed by SDS-PAGE and negative-stain EM before cryo-EM grid preparation.

### Expression and purification of A$_{2A}$R-BRIL

For expression, thermostabilized A$_{2A}$R-BRIL with an N-terminal HA signal sequence and FLAG epitope tag was cloned into a pcDNA™3.1/Zeo(+) vector containing a tetracycline-inducible expression cassette. This construct was transfected into inducible Expi293F cells (Thermo Fisher) using the ExpiFectamine transfection reagent per manufacturer instructions.

For purification, cells were lysed with hypotonic buffer (20 mM HEPES pH 7.5, 1 mM EDTA) supplemented with protease inhibitors (20 μg/mL leupeptin, 160 μg/mL benzamidine). The membrane fraction was solubilized with 20 mM HEPES pH 7.5, 300 mM NaCl, 1% (w/v) lauryl maltose neopentyl glycol (L-MNG, Anatrace), 0.1% cholesteryl hemisuccinate (CHS, Steraloids), protease inhibitors, 5 mM ATP, 2 mM MgCl$_2$, and 1 μM ZM241385 for 1 h at 4 °C. After high-speed centrifugation, the supernatant was affinity purified using M1 anti-FLAG antibody coupled to Sepharose beads. A$_{2A}$R-BRIL bound to M1-beads was washed to gradually decrease detergent and salt concentration and was eluted in 20 mM HEPES pH 7.5, 150 mM NaCl, 0.0075% (w/v) L-MNG, 0.0025% (w/v) glyco-diosgenin (GDN, Anatrace), 0.001% CHS, 1 μM ZM241385, 5 mM EDTA, and 0.2 mg/mL FLAG peptide (Genscript). Eluted A$_{2A}$R-BRIL was concentrated with a

50 kDa MWCO spin concentrator (Millipore) and purified to homogeneity with size-exclusion chromatography, using a Superdex S200 Increase 10/300 GL column (GE Healthcare) equilibrated in 20 mM HEPES pH 7.5, 150 mM NaCl, 0.0075% (w/v) L-MNG, 0.0025% (w/v) GDN, 0.001% CHS, and 1 μM ZM241385. Fractions containing monodisperse A$_{2A}$R-BRIL were pooled, mixed with 2× molar excess of BAG2 fab, and incubated overnight at 4 °C. The next day, the complex was concentrated with a 50 kDa MWCO spin concentrator, and excess fab was removed via size-exclusion chromatography, using a Superdex S200 Increase 10/300 GL column (GE Healthcare) equilibrated in 20 mM HEPES pH 7.5, 150 mM NaCl, 0.00075% (w/v) L-MNG, 0.00025% (w/v) GDN, 0.0001% CHS, and 1 μM ZM241385. The resulting A$_{2A}$R-BRIL BAG2 Fab complex was concentrated with a 50 kDa MWCO spin concentrator to 4.9 mg/mL for the preparation of cryo-EM grids.

### Expression and purification of BAG2 Fab

The BAG2 Fab in pRH2.2 vector was a gift from Kossiakoff lab[23]. This vector was transformed into BL21 Rosetta Escherichia coli cells and grown overnight in Luria Broth supplemented with 50 μg/mL kanamycin shaking at 225 rpm and 37 °C. The next day, the saturated overnight culture was used to inoculate 4 L of Terrific Broth (supplemented with 0.1% glucose, 2 mM MgCl$_2$, and 50 μg/mL kanamycin), and cells were grown shaking at 225 rpm at 37 °C. When cells reached an OD600 = 0.6, expression was induced with the addition of 400 μM IPTG, and the temperature was reduced to 20 °C for 21 h. Cells were harvested by centrifugation and stored in the −80 °C until further use.

For purification, cells were lysed by sonication in PBS, and the soluble fraction was heated in a 60 °C water bath for 30 min. After high-speed centrifugation, the supernatant was affinity purified using Protein A affinity resin (G-Biosciences) according to the manufacturer's protocol. Eluted BAG2 Fab was concentrated with a 10 kDa MWCO spin and further purified with size-exclusion chromatography, using a Superdex S200 Increase 10/300 GL column equilibrated in 20 mM HEPES pH 7.5, 150 mM NaCl. Monodisperse fractions were concentrated and frozen for future use.

### EM sample preparation and data acquisition

Regarding negative-stain EM, 2.5 μl of SMO samples at ~40 μg ml⁻¹ were applied to a glow-discharged Cu grid covered by continuous carbon film, and then stained with 0.75% (w/v) uranyl formate[40]. A Tecnai T12 microscope (ThermoFisher FEI Company) operated at 120 kV was used to analyze these negatively stained grids. Images were recorded at a nominal magnification of 52,000× using an UltraScan 4000 camera (Gatan), corresponding to a pixel size of 2.21 Å on the specimen. To prepare cryo-EM grids of the reconstituted SMO, 3 μl of samples were applied onto a glow-discharged gold grid covered with holey carbon film (Quantifoil, 300 mesh 1.2/1.3) and blotted using a Vitrobot Mark IV (FEI Company) with 3-s blotting time and 100% humidity at 20 °C and plunge frozen in liquid ethane cooled by liquid nitrogen. Grids were imaged with a Titan Krios microscope (ThermoFisher FEI) operated at 300 keV, equipped with a Bio Quantum post-column energy filter with zero-loss energy selection slit set to 20 eV and a K3 camera (Gatan Inc), operating in super-resolution counting mode. Movie stacks were collected using SerialEM[41]. The detailed collecting parameters, including dose rate, total dose and total frames per movie stack, etc. are summarized in the Supplementary Table 1.

To prepare cryo-EM grids of the A$_{2A}$R-BRIL-BAG2 complex, 3 μL of samples at 4.9 mg/mL were added to 300 Mesh 1.2/1.3 R Au Quantifoil grids (glow discharged at 15 mA for 30 seconds with a Pelco easiGlow Glow discharge cleaning system). Grids were blotted with Whatman No. 1 qualitative filter paper in a Vitrobot Mark IV (Thermo Fisher) at 4 °C and 100% humidity for 3 second using a blot

force of 1 prior to plunging into liquid ethane. Images of purified complex were acquired on Titan Krios I at the UCSF Cryo-EM Center for Structural Biology equipped with a BioQuantum K3 Imaging Filter (slit width 20 eV) and a K3 direct electron detector (Gatan) and operating at an acceleration voltage of 300 kV. Images were recorded at a defocus range of −1.0 to −2.0 μm with a nominal magnification of 105 K, resulting in a pixel size of 0.835 Å. Each image was dose-fractionated into 117 movie frames with a total exposure time of 5.9 s, resulting in a total dose of ~67 electrons per Å$^2$. SerialEM was used for data collection.

## Imaging processing

For the datasets of SMO in Salipro system, motion correction and dose weighting of movie stacks were performed on-the-fly using MotionCor2[42]. The contrast transfer function (CTF) was determined using Patch-Based CTF Estimation in cryoSPARC[43]. Particles were picked using a template selected from previous 2D averages followed by reference-free 2D classification (particles binned 4 × 4 during extraction) in cryoSPARC. After removing particles within junk classes, An ab initio 3D reference model was generated using cryoSPARC, which is used for the following 3D classification with a global mask (particles binned 4 × 4) in RELION-3. Particles with bad shape for the sample mSMO-PGS1 and without well-defined 7TM density for the sample mSMO-PGS2 were removed, respectively, and the remaining particles were re-extracted (particles binned 2 × 2) for another round of 3D classification in RELION-3. The particle projections from 2 out of 4 classes for the sample mSMO-PGS1 and 3 out of 8 classes for the sample mSMO-PGS2 were subjected to the 3D classification (Relion options:−tau2_fudge 20 −skip align) with a global or receptor-focused mask, respectively. The remaining particles in one class with well-defined receptor density were retained for 3D refinement in RELION-3. The reported resolution of the reconstruction map was further improved after refinement in RELION-3 using the particle stacks with Salipro disc subtraction. Directional Fourier Shell Correlation (dFSC) curves are calculated as described[44]. Apart from the resolution values reported by RELION, the nominal resolution is also estimated from the averaged FSC using FSC = 0.143 criterion[45]. Local resolution maps were calculated in RELION-3. Conversion of star files from cryoSPARC to RELION-3 was performed using UCSF pyem (https://zenodo.org/record/3576630#.XuczyFVKjIU).

For the dataset of A$_{2A}$R-BRIL-BAG2 complex, a total of 2539 movie stacks were motion corrected and electron-dose weighted using MotionCor213. The CTF parameters were determined, and all subsequent 2D and 3D classifications were performed using cryoSPARC. The initial particle picking was performed by cryoSPARC blob picker. 278,828 particles were selected after several rounds of 2D classification from 1,636,914 particles. The following ab initio reconstruction, heterogeneous refinement, and nonuniform refinement enable us to reconstruct the 4 Å structure with 135,287 particles, from which we generated the template to auto-pick particles. With processing 2,788,578 particles, the map quality is significantly improved after heterogeneous refinement and nonuniform refinement, finally yielding the 3.7 Å map with 335,610 particles. To further improve the resolution, the final particle sets being used for nonuniform refinement from blob picker and template picker were combined after removing the duplicated particles, the following additional round heterogeneous refinement, RELION-based bayesian polishing, and cisTEM-based manual refinement were performed, yielding the 3.4 Å map with 215,946 particles. The cisTEM-based focused refinement enables the further improvement of map quality in terms of TMs region.

The structure determination mainly rests on cryoSPARC, through which the features of Fab, BRIL, and 7TM domain are distinguishable even from the 2D classification result. Multiple rounds of heterogeneous refinement were performed to deal with the flexibility between A$_{2A}$R-BRIL and Fab. To further improve the map quality, RELION-based bayesian polishing and cisTEM-based manual refinement were performed to better determine particles' Euler angles,

## Model building

Atomic models of SMO (PDB ID: 6O3C), PGS (PDB ID: 5U09), and A$_{2A}$R-BRIL (PDB ID: 4EIY) were initially docked into the postprocessed density maps with UCSF Chimera[46] and manually adjusted in Coot[47]. All models were refined over multiple rounds using the module 'phenix.real_space_refine'[48] in PHENIX and ISOLDE[49] implemented in ChimeraX[50]. The quality of all refined models was assessed using the 'comprehensive model validation' function in PHENIX and wwPDB validation server[51]. UCSF Chimera and ChimeraX were used to make figures.

## Reporting summary

Further information on research design is available in the Nature Research Reporting Summary linked to this article.

## Data availability

EM density maps and the related coordinates are deposited to the Electron Microscopy Data Bank (EMDB) and Protein Data Bank (PDB) with access codes EMD-25648 and 7T32 (A$_{2A}$R-BRIL/Fab), EMD-27063 (mSMO-PGS1), EMD-27062 and 8CXO (mSMO-PGS2). All other data are available from corresponding authors upon reasonable request.

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

## Acknowledgements

We thank Dr. Benjamin R. Myers for sharing the initial plasmid of SMO and Dr. Ishan Deshpande for helping with functional experiments. Equipment at UCSF cryo-EM facility was partially supported by National Institutes of Health (NIH) grants (S10OD020054, S10OD021741, and S10OD025881) and is managed by Dr. David Bulkley and Mr. Glenn Gilbert. K.Z. was supported by a Human Frontier Science Program Postdoctoral Fellowship. This work was supported by NIH grants R01GM138992 (A.M.) and R35GM140847 (Y.C.). A.M. acknowledges support from the Pew Charitable Trusts, the Searle Scholars Program, the Vallee Foundation, and the Mallinckrodt Foundation. Y.C. is an investigator of Howard Hughes Medical Institute.

## Author contributions

K.Z. performed all studies on SMO, N.H. purified A$_{2A}$R-BRIL sample and helped with cryo-EM experiments, H.W. performed cryo-EM study on A$_{2A}$R-BRIL. A.M. and Y.C. supervised the study. All authors interpreted data and participated in manuscript preparation.

## Competing interests

A.M. is a consultant for and stockholder in Septerna, Inc. Y.C. is an advisor for Shuimu BioSciences Ltd. The remaining authors declare no competing interests.
