## [Peer Review File · Nature Communications]

Fusion protein strategies for cryo-EM study of G protein-coupled receptorsREVIEWER COMMENTS

Reviewer #1 (Remarks to the Author):

Zhang et al have studied two different fusion partners, BRIL and PGS, for their utility in the structure determination by cryo-EM of two different GPCRs in an inactive state. This is a highly topical and important area as it potentially opens the door to an increased rate in the structure determination of inactive-state GPCRs, similar to what has been observed for active state structures over the last 5 years.

The two targets chosen for study already had structures determined by X-ray crystallography, which allowed the choice of whether a rigid fusion was present and how it was formed. Three important points were discovered in the utility of cryo-EM for structure determination of fusion proteins.

1. BRIL on its own was insufficient for structure determination of the GPCR, but adding an Fab to the BRIL made it possible
2. BRIL fused rigidly via one helix was too flexible for structure determination
3. PGS fused via one helix was successful due to additional stabilising interactions between a loop in PGS and the intracellular surface of the receptor.

These findings will help further rational design of fusions that will add to the strategies already available for cryo-EM structure determination of inactive-state GPCRs.

The manuscript is very well written and I only have a few comments for the authors to address.

1. Line 89; could you please add that the unliganded SMO is in an inactive state, for clarity.
2. Line 124; the density seems to fit phosphatidylserine and that is consistent with the published MS data. However, the MS experiment was not performed on the sample you used for structure determination and different preparations could easily have different amounts/types of lipids present. It is probably better to add putative/potential in relation to the phosphatidylserine, both here and throughout the text (e.g. line 307).
3. Line 163; you anticipated that a single extended helix may be sufficient to maintain a rigid attachment, but this is in conflict with your conclusion on the previous page about BRIL where you say it is not sufficient. This is contradictory and may be confusing, and therefore would benefit with being changed.
4. Line 199. Could you give the surface areas in contact between PGS and the respective receptors. Is the same loop from PGS always involved? Are the same residues in Smo, H1R and GLP-1R all in contact with PGS? Is the orientation of PGS with all the receptors the same? I tried to answer these questions by looking at Fig 4, but it was not immediately obvious from the figures. It would be nice to see equivalent figures to panels a & b for the other receptors. Also how about having a view of the intracellular surface of the aligned receptors with atoms making contact with PGS as spheres?
5. Line 202; it would be helpful again to say that SMO is in an inactive conformation.
6. Line 248; it perhaps would be clearer if this was described as a 'sterol within the 7TM bundle', rather than just a '7TM sterol'.
7. Lines 259-262; I agree with your rationale, but according to your methods you did not add cholesterol to your purification buffers, you added cholesteryl hemisuccinate (CHS). You should therefore model in CHS and not cholesterol as they are chemically different.
8. Lines 276-278; Please add a few references to what is the current model for activation.
9. The discussion focuses only on creating a rigid linker between the fusion protein and the GPCR, but there are two other aspects that could also affect resolution, the detergent used and the ligand. It would be nice if you can comment on these aspects (either here or elsewhere). So for example, would you have got the same resolution structure of A2aR if you had tried to determine the structure of the unliganded receptor? Why did you use Salipro for SMO; would you have got the same resolution using e.g. LMNG/CHS or digitonin? What is the resolution at the orthosteric binding pocket for the A2aR structure? It is noticeable that the resolution deteriorates the further away from the fusion partner you are, and of course this is the key area for drug discovery and is of keen interest to medicinal chemists. Would this strategy be useful for low affinity ligands which would be inevitable in the hit-to-lead phase of a drug discovery project?

10. Line 360; do you mean yellow ribbon instead of pink?

11. Fig. 2; in panels c/d the structure (pink) is referred to as a model, but you use the same word in the text for the AlphaFold derived model. Perhaps use the word structure instead of model in the figure and legend to avoid confusion?

12. Fig S10; In all the six panels to the right, the density for the CRD is a filled-in shape. For the SMO-apo structures it would be clearer if it was left white with a dashed line, as you did not see any density.

Reviewer #2 (Remarks to the Author):

This manuscript by Zhang et al, tackles the important issue of approaches that can extend the application of cryo-EM to broad study of apo and inhibitor bound GPCRs. An understanding of the utility and limitations of different strategies for success would be of high value to the field. The work looks at 2 fusion protein approaches, replacing ICL3, that have previously been applied for GPCR crystallography. They showed that they could get robust maps for an A2AR-BRIL fusion and SMO-PGS and put forward hypotheses on where these approaches could be successful for other receptors. However, they do not really perform any systematic assessments that would provide guidance on utility beyond “you could try fusions with these proteins”. Very substantial additional work would be required to move this beyond a limited observational study.

Specific comments:

Para starting on line 68: There is relevant literature that is not cited.

The CGRPR in apo and peptide-bound forms (Josephs et al, Science 2021) that was solved without fusions (but has an ECD that extends beyond the membrane/micelle) is relevant to the size and mobility of extra-membranous fiducial markers that could support reconstructions.

Assuming that BioRxiv papers are citable in Nat Commun – the Robertson et al. manuscript on kappa-OR ICL3 fusion with Nb6 as an approach to inactive-state GPCR cryo-EM structure (<https://doi.org/10.1101/2021.11.02.466983>) is relevant to discussion on different approaches...

Lines 83-85 – the above-described papers are very relevant to the questions being proposed...

Lines 89-91 – the conclusion is markedly overstated relative to the evidence provided

Line 107 – it would be useful to include the size/Mw of BRIL (and later PGS)

For the A2AR, our laboratory has also used this specific construct as a proof-of-principle for use of BRIL fusion and anti-BRIL Fab as an approach. We also tried this with and without the anti-Fab Nb used in the Fzd5 structure. The Fab itself has a degree of flexibility that is attenuated when using the Nb and this can improve resolution. We achieved ~2.8 Å but our data is essentially equivalent to that presented in the manuscript. Even using a slightly different detergent mix for solubilisation, we also see excellent density for the segment assigned to phosphatidylserine, confirming this observation. However, the conclusion w.r.t. the utility is overstated from using this specific A2AR construct. This construct has extensive mutations that are designed to limit conformational flexibility within the receptor (and thus increase thermostability). I am aware of other work where BRIL is rigidly extended from ICL3 that enables okay resolution of the BRIL/Fab/ICL3 but where the resolution in other parts of the receptor are poor – including in the extracellular half where ligand binding occurs.

The authors need to repeat the work using a non-stabilised form of the A2AR to test the extent to which the engineered receptor was a contributing factor in the achieved outcome.

Line 145 “construct has only a single extended helix between TM5 and the N-terminus of SMO” – I don't know what the authors mean by this statement, it does not appear to make structural sense...

Lines 149-150 – As noted above in Fzd5 the complex included a Nb to further stabilise the Fab. Was this also tried? Moreover, as noted by the authors, the group who solved the Fzd5 structure had to try multiple sites of BRIL insertion to get the approach to work. Without exploring different insertions it is

not clear how the authors can conclude that a BRIL fusion would definitely not work in the case of SMO. In the context of SMO the construct was developed for crystallography and thus the insertion site may not be ideal for cryo-EM.

On SMO-BRIL – there are no methods for this work. Was this construct solubilised and reconstituted in saposin nanodiscs (as per the SMO-PGS constructs)? The way the manuscript is currently written, it implies that the nanodisc approach was only applied to the PGS fusions.

There is a requirement for additional work to compare the different reconstitution systems on the potential success of the different fusion approaches. Currently, it is not possible to understand the relative importance of this to how either fusion strategy should be employed for other receptors.

Lines 187-188 – This conclusion is not robustly supported by the available data. Very few constructs were designed and tested (only one for BRIL, only 2 for PGS), reconstitution systems do not appear to have been controlled for. In regard to mSMO-PGS1, I was not convinced that the map had sufficient resolution to be confident that the fusion had fully extended helix versus a similar TM5 helix to that present in mSMO-PGS2 (having placed the PGS2 model into the map for the mSMO-PGS1).

Is the structure of the PGS loop that contacts SMO (in mSMO-PGS2) the same in all PGS structures, or is this specific to the way it interacts with SMO?

Lines 199-207. This speculation goes beyond what is currently supported by the data. There is no evidence that, just because other receptors have hydrophobic patches on the intracellular face, PGS will form equivalent stable interactions enabling robust map reconstructions. The PGS2 interactions were not predicted by AF2, and in the PGS1 construct these interactions don't occur. If the authors want to promote PGS fusions as a general strategy, they need to demonstrate that it works on other receptors (I would like to see at least 2 examples but minimally 1 other – e.g. the GLP-1R given this is one of their proposed equivalents).

Lines 210-214 – as per my comments above, I was not convinced that PGS1 necessarily had an extended helix (beyond that seen in PGS2). As such, I don't think this conclusion is warranted based on the available data.

Lines 255-256 – while this is likely okay, given the low resolution of the map, I would advise a caveat about the limitations of low-resolution maps for confident modelling.

Lines 265-271 – while there is density in this region of the map it is not well resolved and likewise the density of the surrounding receptor is of limited resolution. Additional evidence (e.g. mutagenesis) that supports the speculation (beyond the weak density) would be advisable if the authors wish to make claims about the new map supporting mechanism.

Lines 293-296 – while the data are potentially consistent with this, they have not performed like for like comparisons, have not included the anti-Fab Nb, etc., so the conclusion per se is not valid.

Lines 301-302 – IMO there needs to be better evidence that the extended helix was present.

Lines 304-305 - How is this a strategy per se? Can this be designed? The solved structure did not match the AF2 prediction for example. They need to demonstrate an actual strategy that can be used. Also, it is not clear the extent to which (or not) the different reconstitution system impacts on stability of the fusion with the receptor etc.

Lines 314-329 – There needs to be a better discussion of the available literature and how that impacts on potential strategies (e.g. the Josephs et al, 2021 and Roberstson et al, BioRxiv papers).

Minor:

Line 22. "Although the similar strategy..." phrasing needs correction

Line 26 (and elsewhere). "3.4Å" should be "3.4 Å" etc

Line 40 "GPCR" should be "GPCRs"

Line 49-50. "either bound to" rephrase to "bound to either"

Line 60. I would suggest changing from the absolute term "regions exist" to "regions often exist" as this is not an absolute

Line 61 "outside of 7TM domain" phrasing needs correction

Line 67 "can facilitate accurate image alignment" – consider adding "if rigidly associated"

Patrick Sexton

We thank both reviewers for their time and efforts in providing us their detailed feedbacks and their suggestions and comments concerning our manuscript. Following their suggestions and comments, we have now revised our manuscript. Our detailed point-to-point rebuttals are listed below. Our responses are color in blue

Reviewer #1 (Remarks to the Author):

Zhang et al have studied two different fusion partners, BRIL and PGS, for their utility in the structure determination by cryo-EM of two different GPCRs in an inactive state. This is a highly topical and important area as it potentially opens the door to an increased rate in the structure determination of inactive-state GPCRs, similar to what has been observed for active state structures over the last 5 years.

The two targets chosen for study already had structures determined by X-ray crystallography, which allowed the choice of whether a rigid fusion was present and how it was formed. Three important points were discovered in the utility of cryo-EM for structure determination of fusion proteins.

1. BRIL on its own was insufficient for structure determination of the GPCR, but adding an Fab to the BRIL made it possible
2. BRIL fused rigidly via one helix was too flexible for structure determination
3. PGS fused via one helix was successful due to additional stabilizing interactions between a loop in PGS and the intracellular surface of the receptor.

These findings will help further rational design of fusions that will add to the strategies already available for cryo-EM structure determination of inactive-state GPCRs.

We appreciate the positive comment and the brief summaries of our findings.

The manuscript is very well written and I only have a few comments for the authors to address.

1. Line 89; could you please add that the unliganded SMO is in an inactive state, for clarity.

Revised.

2. Line 124; the density seems to fit phosphatidylserine and that is consistent with the published MS data. However, the MS experiment was not performed on the sample you used for structure determination and different preparations could easily have different amounts/types of lipids present. It is probably better to add putative/potential in relation to the phosphatidylserine, both here and throughout the text (e.g. line 307).

Fully agreed. We now revised the text.

3. Line 163; you anticipated that a single extended helix may be sufficient to maintain a rigid attachment, but this is in conflict with your conclusion on the previous page about BRIL where you say it is not sufficient. This is contradictory and may be confusing, and therefore would benefit with being changed.

We thank the reviewer for pointing this out. Indeed, the original argument was not presented clearly, making it hard to follow. When BRIL is added to SMO as a fusion protein, either without or with an anti-BRIL Fab, we failed to determine a high-resolution structure of SMO. One possible reason is that a single helix can connect BRIL to SMO with sufficient rigidity, but BRIL itself without an anti-BRIL Fab is too small, as tested in the case of A_{2A}R structure. Another possible reason is that a single helix cannot connect BRIL to SMO with sufficient rigidity and adding a Fab may amplify any flexibility further. Regardless, the conclusion is that single helix BRIL fusion is insufficient for high resolution structure determination, as stated in the section title. We now revised the last sentence of the first result section to emphasize this point.

Our initial hypothesis was that a PGS domain alone is of sufficient size to drive alignment, and thus, a single helix may be alright to facilitate structure determination. But our tests show that a single helix is still insufficient to drive image alignment. We now revised the manuscript to make this rationale clear.

4. Line 199. Could you give the surface areas in contact between PGS and the respective receptors. Is the same loop from PGS always involved? Are the same residues in Smo, H1R and GLP-1R all in contact with PGS? Is the orientation of PGS with all the receptors the same? I tried to answer these questions by looking at Fig 4, but it was not immediately obvious from the figures. It would be nice to see equivalent figures to panels a & b for the other receptors. Also how about having a view of the intracellular surface of the aligned receptors with atoms making contact with PGS as spheres?

Among these three receptors, the only available structure is the SMO fused with PGS presented in this work. During the revision, we found two crystal structures, Melanocortin-4 Receptor (MC4R) and MT₁, both were determined with PGS inserted as a fusion protein to mediate crystal contact. In comparison, the interactions between PGS with MC4R or MT₁ are very similar to that observed in cryo-EM structure of mSMO-PGS2. We now calculated the buried surface areas in contact between PGS and the SMO, together with other two receptors (MC4R and MT₁), as presented in our revised main Figure 3.

In addition, structures of H1R and GLP-1R both have a similar hydrophobic surface areas which could potentially make contacts with PGS, if a PGS is inserted as a fusion protein in the similar manner as presented in our revised Supplementary Figure 9.

Having two examples of PGS fusion protein forming similar hydrophobic interaction with the receptor further suggests that PGS fusion strategy has a potential of being an approach to facilitate structural studies of other GPCRs in the inactive state.

5. Line 202; it would be helpful again to say that SMO is in an inactive conformation.

Revised.

6. Line 248; it perhaps would be clearer if this was described as a 'sterol within the 7TM bundle', rather than just a '7TM sterol'.

Revised.

7. Lines 259-262; I agree with your rationale, but according to your methods you did not add cholesterol to your purification buffers, you added cholesteryl hemisuccinate (CHS). You should therefore model in CHS and not cholesterol as they are chemically different.

We did add cholesterol during the salipro reconstitution. Considering there is no CHS observed in other structures of purified SMO, we putatively model cholesterol into the density map. We further emphasized this in the method section.

8. Lines 276-278; Please add a few references to what is the current model for activation.

References added.

9. The discussion focuses only on creating a rigid linker between the fusion protein and the GPCR, but there are two other aspects that could also affect resolution, the detergent used and the ligand. It would be nice if you can comment on these aspects (either here or elsewhere).

We thank the reviewer for raising these interesting points. Indeed, detergent or lipid and with and without ligand could be two major factors that could influence the achievable resolution. We did not attempt to compare and characterize the influence of detergent vs. lipid environment on the achievable resolution. The main reason we use saposin system for mSMO is to visualize how lipids, particularly cholesterol or its derivative, bind to SMO. Therefore, we did not attempt to characterize the structure of A2aR in detergent versus in lipid environment.

So for example, would you have got the same resolution structure of A2aR if you had tried to determine the structure of the unliganded receptor?

It is known or has been suggested that many GPCRs, such as A2aR, in unliganded apo state are dynamic (such as stated in <https://doi.org/10.1016/j.cell.2013.01.008> and <https://doi-org.ucsf.idm.oclc.org/10.1038/nature17668>). We thus did not attempt determine structure of A2aR in unliganded state. Further, the purpose of fusion strategy is not to solve the conformational heterogeneity caused by protein dynamics. We now added this point in the discussion.

Why did you use Salipro for SMO;

Back to the time when we first developed Salipro approach, we envisioned applying it for small membrane protein without large soluble domains, such as GPCR, considering that Salipro has much smaller unstructured mass than nanodisc. Furthermore, for SMO, a biological goal of this study is to understand cholesterol regulation of SMO. Thus, we chose to use Salipro in this specific case.

would you have got the same resolution using e.g. LMNG/CHS or digitonin?

We thank the reviewer for pointing this out and, indeed, we agree that characterizing such influence of detergent versus lipid environment (nanodisc or Salipro) would be very meaningful. However, it takes a lot of more efforts to thoroughly characterize the influences of detergent versus lipid environment. And the level of influence on the achievable resolution may be case by case without a general guideline. Furthermore, our main focus of this manuscript was to evaluate the different fusion strategies for structure determination of GPCRs in inactive state. Thus, we did not pursue to characterize the influence of detergent vs. lipid environment.

What is the resolution at the orthosteric binding pocket for the A2aR structure?

Based on the local resolution map, the resolution around the orthosteric binding pocket is 3.4-3.6 Å in the A2aR structure (see below).

It is noticeable that the resolution deteriorates the further away from the fusion partner you are, and of course this is the key area for drug discovery and is of keen interest to medicinal chemists.

Indeed, this is the case. This is likely related to the conformational heterogeneity caused by protein dynamics. To improve the resolution of the part distance from the fusion partner, it will require further methodological improvement, most likely better image processing strategies. It is beyond the current study.

Would this strategy be useful for low affinity ligands which would be inevitable in the hit-to-lead phase of a drug discovery project?

This is a very interesting point. At this stage, it is unclear. We envision that determining structure of GPCR with low affinity ligands requires some further innovative efforts.

10. Line 360; do you mean yellow ribbon instead of pink?

Revised.

11. Fig. 2; in panels c/d the structure (pink) is referred to as a model, but you use the same word in the text for the AlphaFold derived model. Perhaps use the word structure instead of model in the figure and legend to avoid confusion?

Revised.

12. Fig S10; In all the six panels to the right, the density for the CRD is a filled-in shape. For the SMO-apo structures it would be clearer if it was left white with a dashed line, as you did not see any density.

Revised.

Reviewer #2 (Remarks to the Author):

This manuscript by Zhang et al, tackles the important issue of approaches that can extend the application of cryo-EM to broad study of apo and inhibitor bound GPCRs. An understanding of the utility and limitations of different strategies for success would be of high value to the field. The work looks at 2 fusion protein approaches, replacing ICL3, that have previously been applied for GPCR crystallography. They showed that they could get robust maps for an A2AR-BRIL fusion and SMO-PGS and put forward hypotheses on where these approaches could be successful for other receptors. However, they do not really perform any systematic assessments that would provide guidance on utility beyond “you could try fusions with these proteins”. Very substantial additional work would be required to move this beyond a limited observational study.

We appreciate the time and effort of Dr. Patrick Sexton. However, with respect, we disagree with this comment. Dr. Sexton failed to recognize the two major findings of this study. First, the **chance of success** is higher if the fusion protein is inserted with two extended helices instead of one. Second, PGS is another option than BRIL that could **potentially** facilitate single particle cryo-EM study of GPCRs in inactive state. We do not intend to present PGS fusion strategy as an approach that **will** work for all GPCR. Instead, we present it as an alternative approach to BRIL fusion strategy.

In the following, we will respond to all comments point-by-point.

Specific comments:

Para starting on line 68: There is relevant literature that is not cited.

The CGRPR in apo and peptide-bound forms (Josephs et al, Science 2021) that was solved without fusions (but has an ECD that extends beyond the membrane/micelle) is

relevant to the size and mobility of extra-membranous fiducial markers that could support reconstructions.

The goal of our study is to explore the possibility of studying GPCRs that without a rigidly attached ECD. Nonetheless, we now added these references.

Assuming that BioRxiv papers are citable in Nat Commun – the Robertson et al. manuscript on kappa-OR ICL3 fusion with Nb6 as an approach to inactive-state GPCR cryo-EM structure (<https://doi.org/10.1101/2021.11.02.466983>) is relevant to discussion on different approaches...

Although bioRxiv papers are not required to be cited, per journal rule, it is citable. We now cited the paper in the revised manuscript.

Lines 83-85 – the above-described papers are very relevant to the questions being proposed...

The bioRxiv paper mentioned is now cited, see above.

Lines 89-91 – the conclusion is markedly overstated relative to the evidence provided

With respect, we disagree. No previous study, at least prior to our initial submission of this manuscript, characterized the significance of having one or two extended helices. Our study DOES provide guideline for future studies to consider.

Line 107 – it would be useful to include the size/Mw of BRIL (and later PGS)

We now added the molecular weight of BRIL.

For the A2AR, our laboratory has also used this specific construct as a proof-of-principle for use of BRIL fusion and anti-BRIL Fab as an approach. We also tried this with and without the anti-Fab Nb used in the Fzd5 structure. The Fab itself has a degree of flexibility that is attenuated when using the Nb and this can improve resolution. We achieved ~2.8 Å but our data is essentially equivalent to that presented in the manuscript. Even using a slightly different detergent mix for solubilisation, we also see excellent density for the segment assigned to phosphatidylserine, confirming this observation.

We are glad to hear that you also obtained similar results. This comment is hardly a criticize but a confirmation of our result. And we appreciate that.

However, the conclusion w.r.t. the utility is overstated from using this specific A2AR construct. This construct has extensive mutations that are designed to limit conformational flexibility within the receptor (and thus increase thermostability). I am aware of other work where BRIL is rigidly extended from ICL3 that enables okay resolution of the BRIL/Fab/ICL3 but where the resolution in other parts of the receptor

are poor – including in the extracellular half where ligand binding occurs.

The authors need to repeat the work using a non-stabilised form of the A_{2A}R to test the extent to which the engineered receptor was a contributing factor in the achieved outcome.

With respect, we disagree. We do not claim that fusion strategy can also stabilize the conformational dynamics of the protein in unliganded apo state. For this reason, we did state in the manuscript that the A_{2A}R is stabilized by a high-affinity antagonist as well as thermostabilizing mutations. Instead, our study aims only to demonstrate that, as a fusion insertion, BRIL requires two rigid connections to achieve high-resolution structure determination. For this purpose, it is advantageous to use the stabilized form of A_{2A}R so that we can circumvent the badly behaved sample and focus on exploring the fusion protein strategies. We now add a sentence to emphasize this point in the result section, and sentences in the discussion to emphasize this point. The results presented in this study would help further rational design of fusions as Reviewer 1 pointed out.

Line 145 “construct has only a single extended helix between TM5 and the N-terminus of SMO” – I don’t know what the authors mean by this statement, it does not appear to make structural sense...

We apologize for the mistake, and thankful to the Dr. Sexton for pointing this out. We meant ‘between TM5 of SMO and N-terminus of BRIL.’ We now revised the text.

Lines 149-150 – As noted above in Fzd5 the complex included a Nb to further stabilise the Fab. Was this also tried?

No, we did not try this. With respect, there is no need to try it. Certainly, an Nb binding to the elbow of a Fab further stabilize a Fab’s constant domain relative to the flexible domain, thus can potentially improve the resolution of the constant domain. As our focus is on the target protein, it is not a major concern if the constant domain of a Fab is not well resolved. As demonstrated in the A_{2A}R-BRIL/Fab case, we reached good resolution without using this nanobody. Our previous studies and hands-on experience as well as studies of many others demonstrated that a rigidly bound Fab with a non-rigid elbow can still facilitate high-resolution structure studies.

Moreover, as noted by the authors, the group who solved the Fzd5 structure had to try multiple sites of BRIL insertion to get the approach to work. Without exploring different insertions it is not clear how the authors can conclude that a BRIL fusion would definitely not work in the case of SMO. In the context of SMO the construct was developed for crystallography and thus the insertion site may not be ideal for cryo-EM.

The reviewer asks the question will BRIL ever work as a fusion for SMO. As shown in the Supplementary Figure 1, we can tell from the crystal structures of Fzd5 and SMO that the orientations of TM5 and TM6 is quite different. Indeed, one can explore many different insertions at different sites to optimize the linkers between receptor and BRIL so that the

BRIL can be used as a fiducial marker to facilitate structure determination of SMO. This is indeed mentioned in the text that it required significant engineering efforts to make Fzd5 work, and cited the reference. And we did not claim that “a BRIL fusion would definitively NOT work in the case of SMO”.

However, the question we asked is if a **single helix extension at one linker site** when the other fusion link is not necessarily grafted with a short/rigid linker, which is much easier to accomplish, is sufficient to guide image alignment. To answer this question, we do not need to exhaust all possible BRIL insertions before exploring an alternative approach.

With respect, the purpose of our study was NOT to demonstrate that the BRIL will never work, which is pointless and would require endless negative control. Rather, our purpose is to find a way to avoid extensive efforts of engineering the link to make it work.

On SMO-BRIL – there are no methods for this work. Was this construct solubilised and reconstituted in saposin nanodiscs (as per the SMO-PGS constructs)? The way the manuscript is currently written, it implies that the nanodisc approach was only applied to the PGS fusions.

We apologize for not clearly stated, we have also reconstituted SMO-BRIL into saposin nanodiscs. We revised our Method part to make it clearer.

There is a requirement for additional work to compare the different reconstitution systems on the potential success of the different fusion approaches. Currently, it is not possible to understand the relative importance of this to how either fusion strategy should be employed for other receptors.

Again, with respect, we disagree. There is no need to compare different reconstitution systems, such as saposin or nanodisc, as our goal is not to distinguish which lipid reconstitution system works better, but simply like to have the SMO in a lipidic environment with cholesterol added.

Lines 187-188 – This conclusion is not robustly supported by the available data. Very few constructs were designed and tested (only one for BRIL, only 2 for PGS),

With respect, we disagree. Again, it is unnecessary to perform an exhausted search for an example in which a single helical extension connecting the fusion, either BRIL or PGS, may work, when there is clearly a better strategy that works.

reconstitution systems do not appear to have been controlled for.

With respect, it is not clear what does Dr. Sexton mean. Both SMO-BRIL and SMO-PGS are reconstituted in saposin nanoparticle (Salipro). It is not clear what “control” experiments does Dr. Sexton suggest?

Nonetheless, we added SEC profiles of SMO-BRIL and SMO-PGS reconstituted into saposin and 2D averages of SMO-BRIL/SMO-BRIL-Fab in the revised Supplementary Figure 5.

In regard to mSMO-PGS1, I was not convinced that the map had sufficient resolution to be confident that the fusion had fully extended helix versus a similar TM5 helix to that present in mSMO-PGS2 (having placed the PGS2 model into the map for the mSMO-PGS1).

With respect and our apology, we are confused about this statement. We try to address this comment based on what we think the reviewer meant.

First, the atomic models of both mSMO-PGS1 and mSMO-PGS2 were predicted by using alphaFold2 (AF2). We did not place the PGS2 model into the map for the mSMO-PGS1. The confusion may come from the labels, both models have an extension _AF2, which means AlphaFold2.

Second, we docked the predicted model of mSMO-PGS1 as a rigid body in the density map of mSMO-PGS1. The model, including the extended helix, fits the density well as a rigid body. However, given the limited resolution of ~6.7Å achieved, which is sufficient to resolve helices, we concluded that the single extended helix from TM6 (we assume the reviewer meant TM6) may not be sufficiently rigid. Otherwise, a sufficiently rigidly attached PGS fusion should be able to drive image aligning to yield a better resolution.

Is the structure of the PGS loop that contacts SMO (in mSMO-PGS2) the same in all PGS structures, or is this specific to the way it interacts with SMO?

By comparing our mSMO-PGS2 structure with structures of other two receptors (MC4R, PDB 6W25 and MT₁, PDB 6ME2), their PGS parts are structurally similar. As presented in our revised main Figure 3, the fusion protein PGS rotated with respect to the receptor.

Lines 199-207. This speculation goes beyond what is currently supported by the data. There is no evidence that, just because other receptors have hydrophobic patches on the intracellular face, PGS will form equivalent stable interactions enabling robust map reconstructions. The PGS2 interactions were not predicted by AF2, and in the PGS1 construct these interactions don't occur.

Indeed, we do not have other cryo-EM structures of GPCRs using the same fusion design as the PGS2 design shown here. Considering that PGS fusion is also used in crystallizing GPCRs, during the revision, we compared our cryo-EM structure of mSMO-PGS2 with some crystal structures. There are several GPCR crystal structures, such as MC4R and MT₁, in which PGS form a similar hydrophobic interaction with the GPCRs being crystallized. In these two structures, the PGS fusion does not have an extended helix from TM6. We now included these two examples in the revised manuscript.

If the authors want to promote PGS fusions as a general strategy, they need to demonstrate that it works on other receptors (I would like to see at least 2 examples but minimally 1 other – e.g. the GLP-1R given this is one of their proposed equivalents).

We appreciate this comment. Indeed, we attempted to craft a PGS fusion to GLP-1R in the same way as in the mSMO-PGS2. However, it requires extended efforts to produce sufficient protein by using our overexpression system. Further, the purified apo GLP-1R was not stable and fusion strategy by its own does not improve protein stability. Concerning that a prolonged time required to complete this experiment would diminish the significance of our work, and that we now added examples from existing crystal structures, we will leave this experiment as follow up studies.

Lines 210-214 – as per my comments above, I was not convinced that PGS1 necessarily had an extended helix (beyond that seen in PGS2). As such, I don't think this conclusion is warranted based on the available data.

With respect, we disagree, same as our response to the comment above.

Lines 255-256 – while this is likely okay, given the low resolution of the map, I would advise a caveat about the limitations of low-resolution maps for confident modelling.

To clarify the map quality for side-chain modeling in this region, we added the new panel e into the revised Supplementary Fig. 8.

Lines 265-271 – while there is density in this region of the map it is not well resolved and likewise the density of the surrounding receptor is of limited resolution. Additional evidence (e.g. mutagenesis) that supports the speculation (beyond the weak density) would be advisable if the authors wish to make claims about the new map supporting mechanism.

We appreciate this suggestion. As shown in Figure 4f, the side chain densities of these named residues are resolved in our density map. No other lipids fit to the shape of this density. We did mutagenesis, but the results are not informative. The main reason is that the assay we used detects the SMO activation but cannot easily test the functional consequences of this specific cholesterol binding in an inactive state. To this end, we did state in the text that our interpretation is speculative.

Lines 293-296 – while the data are potentially consistent with this, they have not performed like for like comparisons, have not included the anti-Fab Nb, etc., so the conclusion per se is not valid.

With respect, we disagree. As responded above, an anti-Fab Nb is unlikely to change the conclusion.

Lines 301-302 – IMO there needs to be better evidence that the extended helix was present.

With respect, we disagree. The resolution of the SMO-PGS1 is sufficient to show that the map matches well with the predicted model.

Lines 304-305 - How is this a strategy per se? Can this be designed? The solved structure did not match the AF2 prediction for example. They need to demonstrate an actual strategy that can be used. Also, it is not clear the extent to which (or not) the different reconstitution system impacts on stability of the fusion with the receptor etc.

With respect, we disagree. We surmise that the linkers between TM5/6 of receptors and PGS should be relatively long in order to allow the formation of hydrophobic interactions between receptors and PGS. As stated above, we have used the same reconstitution system for all constructs of SMO.

Lines 314-329 – There needs to be a better discussion of the available literature and how that impacts on potential strategies (e.g. the Josephs et al, 2021 and Roberstson et al, BioRxiv papers).

We cited the papers as suggested by the reviewer above and revised discussion part.

Minor:

Line 22. “Although the similar strategy...” phrasing needs correction

Revised.

Line 26 (and elsewhere). “3.4Å” should be “3.4 Å” etc

Revised.

Line 40 “GPCR” should be “GPCRs”

Revised.

Line 49-50. “either bound to” rephrase to “bound to either”

Revised.

Line 60. I would suggest changing from the absolute term “regions exist” to “regions often exist” as this is not an absolute

Revised.

Line 61 “outside of 7TM domain” phrasing needs correction

Revised.

Line 67 “can facilitate accurate image alignment” – consider adding “if rigidly associated”

Revised.

Patrick Sexton

REVIEWERS' COMMENTS

Reviewer #1 (Remarks to the Author):

The authors have made a reasonable job of incorporating the suggested changes into the manuscript. Note that reference to 'cholsterols' in the text should be changed to the singular.

We thank the reviewer for providing us their feedbacks and their suggestions and comments concerning our manuscript. The detailed point-to-point rebuttals are listed below. Our responses are colored in blue.

Reviewer #1 (Remarks to the Author):

The authors have made a reasonable job of incorporating the suggested changes into the manuscript. Note that reference to 'cholsterols' in the text should be changed to the singular.

Revised.